# Pharmacokinetics, pharmacodynamics, and toxicity of a PD-1-targeted IL-15 in cynomolgus monkeys

Changhua Ji[1]*, Bing Kuang[2], Bernard S. Buetow[1], Allison Vitsky[1], Yuanming Xu[3], Tzu-Hsuan Huang[3], Javier Chaparro-Riggers[2], Eugenia Kraynov[2], Diane Matsumoto[1]

**1** Drug Safety Research and Development, Pfizer Inc, San Diego, California, United States of America, **2** Biomedical Design, Pfizer Inc, San Diego, California, United States of America, **3** Cancer Immunology Discovery, Pfizer Inc, San Diego, California, United States of America

\* changhua.ji2@pfizer.com

**Data Availability Statement:** All relevant data are within the manuscript and its Supporting Information files. Additional data can be requested

## Abstract

PF-07209960 is a novel bispecific fusion protein composed of an anti-PD-1 antibody and engineered IL-15 cytokine mutein with reduced binding affinity to its receptors. The pharmacokinetics (PK), pharmacodynamics (PD), and toxicity of PF-07209960 were evaluated following once every other week subcutaneous (SC) or intravenous (IV) administration to cynomolgus monkeys in a repeat-dose PKPD (0.01–0.3 mg/kg/dose) and GLP toxicity study (0.1–3 mg/kg/dose). PF-07209960 showed dose dependent pharmacokinetics with a terminal T1/2 of 8 and 13 hours following IV administration at 0.03 and 0.1 mg/kg, respectively. The clearance is faster than a typical IgG1 antibody. Slightly faster clearance was also observed following the second dose, likely due to increased target pool and formation of anti-drug antibodies (ADA). Despite a high incidence rate of ADA (92%) observed in GLP toxicity study, PD-1 receptor occupancy, IL-15 signaling (STAT5 phosphorylation) and T cell expansion were comparable following the first and second doses. Activation and proliferation of T cells were observed with largest increase in cell numbers found in gamma delta T cells, followed by CD4+ and CD8+ T cells, and then NK cells. Release of cytokines IL-6, IFNγ, and IL-10 were detected, which peaked at 72 hours postdose. There was PF-07209960-related mortality at ≥1 mg/kg. At scheduled necropsy, microscopic findings were generalized mononuclear infiltration in various tissues. Both the no observed adverse effect level (NOAEL) and the highest non severely toxic dose (HNSTD) were determined to be 0.3 mg/kg/dose, which corresponded to mean Cmax and AUC48 values of 1.15 µg/mL and 37.9 µg*h/mL, respectively.

## Introduction

Interleukine (IL)-15 is a potent cytokine that enhances the proliferation and effector functions of IL-2Rβ/γ-expressing lymphocytes, such as NK cells and T cell subtypes (CD8+, CD4+, γδ, and NKT). IL-2 and IL-15 share the IL-2Rβγ receptors but have their own alpha receptors, IL-

by contacting the corresponding author (pending institutional legal approval).

**Funding:** The authors received no specific funding for this work.

**Competing interests:** The authors have declared that no competing interests exist.

2Ra and IL-15Ra [1, 2]. IL-2 preferentially leads to expansion of regulatory T cells, while IL-15 plays an important role in maintaining NK cells and CD8+ memory T cells and shows promise as a new immune-oncology therapy [2–4]. IL-2 is an approved immunotherapy (aldesleukin) that can induce complete and durable responses in patients with metastatic melanoma and renal cell carcinoma [5]. However, this efficacy is achieved at high doses that cause substantial toxicity. IL-15 demonstrated potent anti-tumor efficacy in mouse models [6]; however, recombinant IL-15 clinical efficacy has been limited due to a short half-life and toxicities associated with systemic immune activation [7–10]. Recently, new IL-15 modalities aimed at extending half-life have advanced to the clinic with promising preliminary results as monotherapy, or in combination with PD (programmed death)-1 inhibitors [11–13].

Recombinant human IL-15 (rhIL-15) was evaluated in patients with metastatic melanoma and was shown to induce regression of lung metastases in two patients [3], but overwhelming toxicity and a narrow therapeutic window limited its continued development. Of the IL-2Rβ/γ-expressing lymphocytes, NK cells express the highest level of IL-2Rβ and are highly sensitive to the systemically administered IL-15 [14]. Because intratumoral CD8+ T cell activation and expansion is associated with efficacy both in animal models and in human patients [15–17], targeted delivery of IL-15 to these intratumor T cells appears to be an attractive strategy. PD-1 is expressed on tumor-infiltrating T cells at relatively high levels and is a clinically proven immunotherapy target [18–20]. In order to optimally activate intratumoral PD-1+ CD8+ T cells, an anti-human PD-1 antibody-human IL-15 cytokine fusion protein (PF-07209960) was developed to enable targeted delivery of IL-15 to PD-1-expressing intratumoral CD8+ T cells. This new IL-15 modality is expected to have reduced activity on PD-1-negative NK cells and thus an improved systemic toxicity profile. By using an anti-mouse PD-1 antibody and mouse IL-15 mutein (blunted IL-15 receptor binding affinity) fusion protein surrogate, αmPD1-mIL15m, we have previously demonstrated that PD-1 anchoring of an IL-15 mutein rendered potent activity on intratumoral CD8+ with enhanced anti-tumor immune response in mouse tumor models without causing body weight loss, an indicator of systemic toxicity [21]. This data supported the development of PF-07209960 as a new therapeutic cytokine. Here we report the characterization of pharmacokinetics, pharmacodynamics, and toxicity of PF-07209960 in cynomolgus monkeys.

## Materials and methods

### Production of aPD1-IL-15m fusion protein (PF-07209960)

The production of anti-human PD-1–IL15 mutein fusion protein (αPD1-IL15m) was described previously [21]. PF-77209960 consists of a human PD-1-binding human IgG1 antibody with bispecific knob-into-hole mutations on CH3 (Y349C, T366W, S354C, T366S, L368A, and Y407V) that is linked monovalently to an IL15 variant molecule with mutations (N1G-D30N-E46G-V49R-E64Q) at the C-terminus through a flexible "GGGGSGGGGSGGGG" linker. This molecule utilizes a modified (L234A, L235A, and G237A) human IgG1 Fc region to abrogate Fcγ receptor and C1q binding. The five mutations (N1G, D30N, E46G, V49R, and E64Q) were introduced in IL15 to eliminate IL15Rα binding and reduce IL2Rβ/γ affinity. PF-07209960 was produced in stable CHO cell line and purified from the cell supernatant using Protein A affinity chromatography on a MabSelectSuRe column (GE Lifesciences), ion-exchange chromatography on a Mono S 5/50 GL or Mono S 10/100 GL column (GE Lifesciences) and size exclusion chromatography on a HiLoad 16/600 Superdex 200 prep grade or HiLoad 26/600 Superdex 200 prep grade column (GE Lifesciences).

## Surface plasmon resonance analysis

The binding affinities of anti-PD-1-IL-15 fusion proteins to human IL-15Rα and human IL-2Rβ were measured on a Biacore T200 Surface Plasmon Resonance based biosensor (GE Lifesciences). An anti-human Fc reagent (Goat anti-human IgG Fc-γ specific, SouthernBiotech, Catalog#2014–01) was injected into a Biacore CM4 sensor chip (GE Lifesciences) that had been pre-activated with NHS/EDC (N-ethyl-N′-(3-(dimethylamino)propyl)carbodiimide/N-hydroxysuccinimide). All flow cells were then blocked with 100 mM ethylenediamine. All protein interaction experiments were performed at 37˚C using a running buffer of 10 mM HEPES (4-(2-hydroxyethyl)-1-piperazineethanesulfonic acid), 150 mM NaCl, 0.05% (v/v) Tween-20, pH 7.4, 1 mg/mL BSA (bovine serum albumin). Anti-PD-1-IL-15 fusion proteins were captured via the anti-human Fc (10 ug/ml for 2 min) on the surface of the sensor chip. Different concentrations of purified human IL-15Rα or human IL-2Rβ were injected as analytes. After each analyte injection, dissociation was monitored for 5 minutes followed by regeneration of all flow cells with 75 mM phosphoric acid injected three times for 60 seconds. Buffer cycles were collected for each captured fusion protein for double-referencing purposes [22]. For steady-state affinity analysis, the double-referenced equilibrium binding responses were fit with a 1:1 Langmuir steady-state model using Biacore T200 Evaluation Software version 2.0.

## STAT5 (signal transducer and activator of transcription 5) phosphorylation in human and monkey NK and CD8+ T cells

Fresh PBMCs (peripheral blood mononuclear cells) were resuspended in complete RPMI medium and rested for 2 hrs at 37˚C. Cells were resuspended in serum-free RPMI and added into 96 well U-bottom plates and rested for 10 to 30 min. PF-07209960 was added to cells and incubated at 37˚C for 30 min, followed by addition of 4% PFA (paraformaldehyde) (2% final volume) and incubation for 10 min at room temperature. Cells were washed 2x with FACS (fluorescence-activated cell sorting) buffer and stained with cell surface maker antibodies. After fixation, cells were washed 2x and resuspended in Perm buffer III. The permeabilization step proceeded for 30 min on ice after which cells were stained with antibodies against intracellular markers phosphorylated STAT5 (pSTAT5) (45 min to 1 hr at room temperature). After staining, cells were washed 2x and resuspended in 200 μL of FACS buffer for analysis on a flow cytometer.

## Cynomolgus monkey pharmacokinetics (PK) and pharmacodynamics (PD) study

This study complies with all applicable sections of the Final Rules of the Animal Welfare Act regulations (Code of Federal Regulations, Title 9), the Public Health Service Policy on Humane Care and Use of Laboratory Animals from the Office of Laboratory Animal Welfare, and the Guide for the Care and Use of Laboratory Animals from the National Research Council. The protocol and any amendments or procedures involving the care or use of animals in this study were reviewed and approved by the Testing Facility Institutional Animal Care and Use Committee before the initiation of such procedures. Naïve cynomolgus monkeys of Chinese origin (8 females in total), 2–4 kg body weight and 2–4 years of age, were used in this PKPD study. Monkeys were administered with PF-07209960 every other week for two total doses at 0.01, 0.03, 0.1, or 0.3 mg/kg by SC injections, or at 0.01, 0.03, or 0.1 mg/kg by IV bolus injections. Except for the 0.3 mg/kg dose group which had two monkeys, a single monkey was used for all other dose groups. The dosing of animals was performed at Charles River Laboratories, Reno, NV (Study #20160115). All animals were returned to colony at the end of this study (29-Day

study). PK and PD analyses (pSTAT5 and PD-1 receptor occupancy) were performed by Pfizer BioMedicine Design (BMD) department in La Jolla, CA.

## PD-1 receptor occupancy

Receptor occupancy was quantified using a commercial antibody against human PD-1 (CD279, clone EH12.1, BD Biosciences). This antibody binds to the same epitope as PF-07209960 and was used to detect unoccupied PD-1 on the cell surface. Anti-PD-1 antibody EH12.1 and the cell phenotyping antibodies directed against CD3, -4, -8, and -159a for identifying T cell subsets and NK cells were added to dosed monkey blood samples. After incubation on ice for 60 min, RBCs were lysed by adding BD Pharm Lyse (BD Biosciences). Cell pellets were washed twice with FACS buffer and fixed with 2% paraformaldehyde (PFA), and analyzed on a BD LSRII (BD Biosciences) flow cytometer. Cell surface PD-1 abundance was quantified using PF-07209960, which was detected using a secondary antibody. Predose blood samples were used to create the 100% drug-free (0% occupancy) control signal, and monkey blood samples pulsed with 1 μg/mL of PF-07209960 for saturation of cell surface PD-1 binding sites were used to generate the 0% drug-free (100% occupancy) control signal. The percentages of drug-free PD-1 on T cell or NK cell surface in PF-07209960-dosed monkey blood samples were calculated by using these two controls.

## STAT5 phosphorylation in whole blood

Whole blood samples from each collection timepoint were mixed with BD Phosflow Lyse/Fix Buffer and incubated at 37˚C for 10–15 min. Cells were then washed once with 2 mL of FACS Buffer (PBS containing 0.5% bovine serum albumin) and resuspended in 1 mL of -20˚C, 100% methanol on ice for 30 minutes. The permeabilized samples were then stored at -20˚C in methanol. Samples were aliquoted, washed, and resuspended in 10 uL of FACS buffer. An antibody cocktail (35 uL) containing Brilliant Stain Buffer Plus, pSTAT5 AF647, CD45 BUV395, CD3 FITC, CD4 PE-Cy7, CD8 BV421, and CD159a PE was added to each well and incubated for 30 min at room temperature. All antibodies were purchased from BD Biosciences except for CD159a, which was from Beckman Coulter. Cells were washed 3 times with FACS wash buffer before adding BD Stabilizing Fixative. The samples were analyzed on a BD LSRFortessa X-20 flow cytometer. Data analysis was performed using FACS Diva software. The percentage of cells that were pSTAT5 positive in the AF647 channel from each of the cell populations listed above at each time point in the study was used to demonstrate phosphorylation of STAT5. The percentage of pSTAT5+ cells at time = 0 (day 1 predose) when using the pSTAT5 antibody was used to set a background gate.

## Cynomolgus monkey GLP toxicity study

The dosing of animals was performed at Covance Laborotories Inc., Madison, WI (study #8420678). This study was conducted in accordance with the current guidelines for animal welfare (National Research Council Guide for the Care and Use of Laboratory Animals, 2011; Animal Welfare Act, 1966, as amended in 1970, 1976, 1985, and 1990, and the Animal Welfare Act implementing regulations in Title 9, Code of Federal Regulations, Chapter 1, Subchapter A, Parts 1–3). All procedures in the protocol were in compliance with applicable animal welfare acts and were approved by the local Institutional Animal Care and Use Committee (IACUC). In this study, a total of 15 male and 15 female Mauritius cynomolgus monkeys were used. At the initiation of dosing, body weight ranged from 2 kg to 4.7 kg, and age ranged from 2.5 to 4.5 years old. Pharmacokinetics of PF-07209960, ADA analysis, and immunophenotyping were performed by Pfizer BioMedicine Design (BMD) and DSRD departments. Animals

were acclimated for at least 3 weeks, they were socially housed (up to 3 animals/cage). Certified Primate Diet #5048 (PMI Nutrition International Certified LabDiet®) was provided 1 or 2 times daily. Animals were given various cage-enrichment devices and dietary enrichment. All animals were observed twice daily.

Male and female cynomolgus monkeys were assigned to 5 groups (3 males and 3 females per group). The number of animals included in this GLP toxicity study is consistent with regulatory guideline (ICH S9). Vehicle control or PF-07209960 at 1, 3, 0.3, or 0.1 mg/kg/dose were administered via subcutaneous injection (1 mL/kg dose volume) into the scapular region on Days 1, 15, and 29 of the dosing phase. Assessment of toxicity (included monitoring for suffering) was based on mortality, clinical observations (twice daily), body weights (at least once a week), qualitative food consumption (daily), body temperatures (On Days 1, 2, 3, 7, 14, 15, 16, 17, 18, 21, 28, 29, 30, and 31 of the dosing phase), ophthalmic observations (Once during the acclimation phase (prior to the initiation of dosing) and once during the final 7 days of the dosing phase), electrocardiographic (ECG) measurements (predose and 2 hours postdose on Day 1), and clinical and anatomic pathology. Blood samples were collected for toxicokinetic (TK), anti-drug antibody (ADA), cytokine, and immunophenotyping analyses. The timepoints for TK were predose and approximately 2, 6, 24, 48, 72, 96, 120, 144, and 168 hours postdose on Day 1 (first dose) and Day 15 (second dose), and predose and approximately 2, 6, 24, and 48 hours postdose on Day 29 (third dose).

For scheduled (Day 31) and unscheduled euthanasia, animals were anesthetized with sodium pentobarbital (50 mg/kg) and exsanguinated before necropsy. All animals (6 in total) at 3 mg/kg/dose and 4 out of 6 animals at 1 mg/kg/dose were euthanized early in the dosing phase due to toxicity (10 animals in total). The rest animals administered PF-07209960 (2 animals at 1 mg/kg, 6 animals at 0.1 mg/kg, and 6 animals at 0.3 mg/kg (14 in total)) survived to scheduled euthanasia on Day 31. and General criteria for unscheduled euthanasia include one or more of the following: marked losses in body weight, dehydration that cannot be normalized with fluid therapy, or demonstration of severe dysfunction of one or more organ systems by clinical signs and/or laboratory test results. In this study a combination of hyperthermia, gastrointestinal signs, changes in posture, decreased activity, petechia/bleeding and/or extreme decreases in platelets led to the decision to humanely euthanize affected animals prior to the scheduled euthanasia.

## Histologic evaluations

A full necropsy was conducted on all monkeys. Microscopic examination of formalin-fixed, paraffin-embedded, hematoxylin and eosin-stained sections was performed on a standard comprehensive list of tissues (S1 Table in S1 File). Tissues were examined by a board-certified veterinary pathologist and findings were recorded on a subjective scale as follows: minimal = an inconspicuous change, mild = a noticeable but not prominent change, moderate = a prominent change, marked = a dominant but not overwhelming change, and severe = an overwhelming change. A second board certified veterinary pathologist reviewed the findings and the results reported herein represent the consensus opinion of the two pathologists.

## Clinical pathology and biomarker analysis

Blood samples for hematology were collected at baseline as well as on Days 1, 3, 5 8, 15, 17, 19, 22, 29, and 31. Blood samples for coagulation were collected at baseline as well as on Days 5, 17, 19, and 31. Blood samples for clinical chemistry and C-reactive protein were collected at baseline and Day 31. In addition, blood was collected for unscheduled analysis of hematology,

coagulation, and/or clinical chemistry from individual animals on Days 4, 6, 7, and/or 16. Hematology was performed on a Siemens Advia 2120 Hematology Analyzer (Siemens Health-care Diagnostics, Tarrytown, NY). Coagulation parameters were performed on an Instrumentation Laboratory ACL Top 500 CTS Analyzer (Werfen Life Group, Bedford, MA). Clinical chemistry and C-reactive protein assays were performed on a Roche Cobas 8000 c702 Analyzer (Roche Diagnostics Corporation, Indianapolis, IN).

### Blood immunophenotyping

Blood cells were stained for routine markers using anti lymphocyte monoclonal antibodies with different fluorescent labels. Approximately 100 μL blood or were aliquoted into each tube containing cocktails of antibodies with different fluorescent labels. The red blood cells were lysed with 1–2 mL of FACS lysing solution supplied by Becton Dickinson (BD). Samples not stained for Ki-67 or FoxP3 were washed 2 times, reconstituted in 0.3 mL of Stain Buffer (BD), and kept refrigerated until flow cytometry analysis. Samples stained for intracellular FoxP3 and Ki-67 were incubated in 2ml of FoxP3 Buffer A (BD) at room temperature for 10 mins, washed twice with Stain Buffer, and then incubated with 0.5ml of Buffer C (BD) for 30 minutes at room temperature. After washing, cells were incubated with the Ki-67 and FoxP3 intracellular staining cocktail for approximately 30 minutes at room temperature. Samples were washed with Stain Buffer and reconstituted in 0.3 mL of Stain Buffer and kept refrigerated for up to 72 hours until flow cytometry analysis, which were performed on a BD FACSCanto flow cytometer with BD FACSDiva v8.0.1 software. The percentage of each lymphocyte subset out of total lymphocytes from blood was determined and were used to determine the absolute counts (reported as cells/μL) for each lymphocyte subset based on the absolute lymphocyte counts derived from hematology data.

### Quantification of PF-07209960 in cynomolgus monkeys

A ligand binding assay (LBA) using Gyros 200 CD platform was qualified to measure the concentration of PF-07209960 in monkey PKPD study. PF-07209960 was captured by a biotinylated mouse anti-PF-07209960 antibody on streptavidin-coated beads. The labeled mouse anti-human IL-15 antibody was used for detection in the PF-07209960 specific assay, while labeled goat anti-human IgG antibody was used for detection in the "Total IgG" assay. A LBA using MSD platform was validated for the quantification of PF-07209960 in cynomolgus monkey serum in the GLP study. In this assay, PF-07209960 was captured by a biotinylated mouse anti-PF-07209960 antibody on a streptavidin-coated Multi-Array plate. The bound PF-07209960 was detected with a ruthenium-labeled mouse anti-human IL-15 antibody, and a read buffer containing tripropylamine (TPA) to produce an electrochemiluminescence (ECL) signal employed within the MSD instrument. Sample concentrations were determined by interpolation from standard curves that were fit using a 5-parameter regression model. The range of quantification in 100% serum was 10 to 1280 ng/mL.

### Detection of anti-drug antibodies in cynomolgus monkeys

A bridging LBA was validated to detect the presence of ADA in cynomolgus monkey serum on MSD assay platform. In this method, biotin-labeled PF-07209960 and ruthenium-labeled PF- 07209960 were co-incubated with study samples and controls. Antibodies to PF-07209960 present in the samples must be bound to both the biotin- and ruthenium-labeled versions of PF- 07209960 to be detected in this assay. ADA complexes were captured via the biotinylated PF-07209960 bound to streptavidin-coated MSD Multi-Array plates. Final detection was conducted by using ruthenium-labeled PF-07209960 and a read buffer containing TPA to produce

ECL signal employed within the MSD instrument. Study samples were tested for ADA using a tiered strategy, and conclusions regarding the induction of ADA were made based on the comparison of the pre- and postdose sample results.

### Cytokine measurement

Blood was collected for cytokine analysis from animals on Day 1 (predose), 24, 48, and 96 hours postdose, on Days 15 (predose and 48 hrs postdose) and Day 29. On Day 4 of the dosing phase, blood was collected from all Group 3 (3 mg/kg/dose) animals prior to euthanasia. In addition, on Day 6 of the dosing phase, blood was collected from Animals P0101 and P0102 (Group 2 males [1 mg/kg/dose]) and P0502 (Group 2 female [1 mg/kg/dose]) prior to euthanasia. Blood was collected into serum separator tubes (without anticoagulant), allowed to clot at room temperature for at least 30 minutes prior to centrifugation, and centrifuged within 1 hour of collection. Serum was harvested and stored at -80˚C until use. Cytokines were measured by Covance Laboratories using Bio-plex reagent kit (Bio-Rad) following manufacturer's instructions.

## Results

### Characterization of aPD-1-IL15 PF-07209960

We first characterized the binding properties of PF-07209960 to PD-1 and IL-15 receptors. Monovalent affinity to the receptor IL-2Rγ was not measured due to very weak affinities.

PF-07209960 did not exhibit any binding to human and cyno IL-15Rα at wide range of concentrations tested (up to 3 μM). Abolishing the binding to IL-15Rα is expected to improve the PK of PF-7209960 by preventing its clearance from the circulation through binding to IL-15Rα expressing cells such as endothelial cells. PF-07209960 showed comparable affinity to human and monkey IL-2Rβ ($K_D$ of 4 μM and 2.4 μM, respectively) and PD-1 ($K_D$ of 0.1 nM and 0.07 nM, respectively). The affinities of PF-07209960 to human and cyno FcRn are also similar (1040 nM and 426 nM, respectively). In addition, PF-07209960 exhibited comparable activity in human and monkey lymphocytes. The pSTAT5 assay EC50 values were 1.1 nM and 2.1 nM in human and monkey NK cells, and 0.2 nM and 0.1 nM in human and monkey CD8 + T cells. Therefore, cynomolgus monkey is a pharmacologically relevant species and was used for nonclinical PKPD and toxicity studies.

### Pharmacokinetics of PF-07209960

The pharmacokinetics of PF-07209960 were determined in cynomolgus monkeys following IV administration at 0.01, 0.03 and 0.1 mg/kg on Days 1 and 15, or SC administration at 0.01, 0.03, 0.1 and 0.3 mg/kg on Days 1 and 15 (n = 1/ dose). PF-07209960 concentrations were determined using a PF-07209960 specific assay and a "Total IgG" assay. The PK profiles observed in monkeys utilizing the two assays largely overlapped, which suggested the fused IL-15 cytokine was not more prone to catabolism than the conjugated anti-PD-1 mAb in vivo. Following IV administration, PF-07209960 had rapid clearance (compared to typical mAbs) [23] but the clearance was consistent with cytokine conjugated mAbs (including CEA-IL2v1) [24]. PK was non-linear, with clearance decreasing from 7.5 mL/hr/kg to 2.2 mL/hr/kg as the dose increased from 0.01 to 0.1 mg/kg (Fig 1A and 1B), which is consistent with the profile of target-mediated clearance. The PK also showed accelerated clearance following the second dose. This is likely due to the increased target pool (expansion of T cells and NK cells) and formation of ADA. Following SC administration, the bioavailability was estimated to be ∼40%.

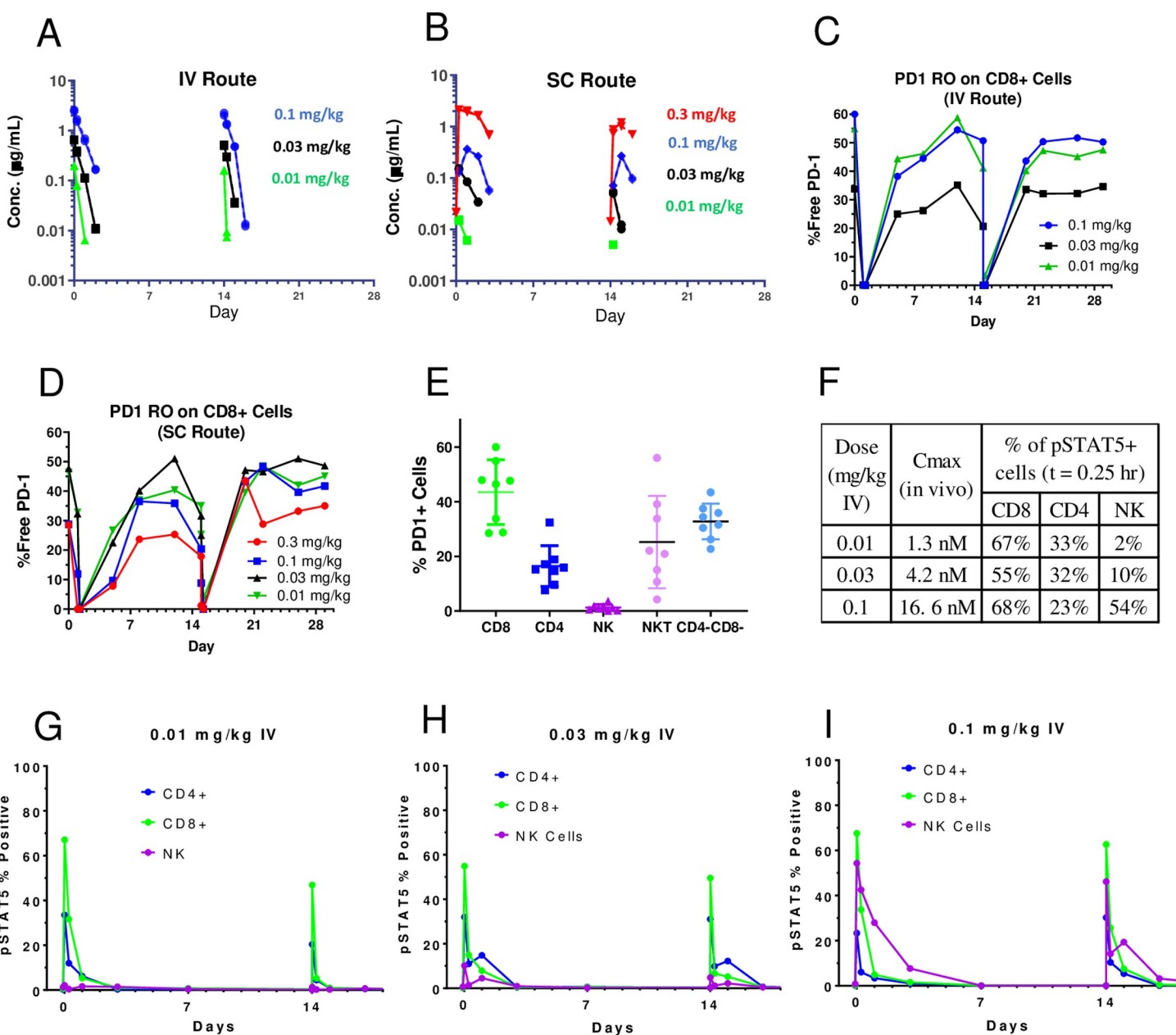

**Fig 1. PK and PD of PF-07209960 in cynomolgus monkeys.** Female monkeys were administered PF-07209960 by intravenous (IV) or subcutaneous (SC) route at 0.01, 0.03, 0.1, or 0.3 mg/kg (SC only) on Days 1 and 15. A single monkey was used per dose group, except for the 0.3 mg/kg SC group where 2 monkeys were used. **A** and **B** represent PK data following IV (**A**) and SC (**B**) dosing. **C** and **D** represent PD-1 receptor occupancy (RO) on CD8+ T cells following IV (**C**) and SC dosing (**D**). **E** shows the percentages of PD-1+ CD4+ T cells, PD-1+ CD8+ T cells, and PD-1+ NK cells in cynomolgus monkey periphery blood, measured on Day 1 predose. **F** lists the Cmax values and the percentages of pSTAT5+ T cell subsets (CD4+ and CD8+) and NK cells at 0.25 hr following IV dosing at 0.01, 0.03, and 0.1 mg/kg. **G**-**H** represent the percentages of pSTAT5+ T cell subsets (CD4+ and CD8+) and NK cells following IV dosing at 0.01 mg/kg (**G**), 0.03 mg/kg (H), and 0.1 mg/kg (**I**).

In a GLP repeat-dose toxicity study in which cynomolgus monkeys were administered PF-07209960 at 0.1, 0.3, 1, 3 mg/kg/dose (SC, once every 2 weeks), PK and ADA were further evaluated. The mean systemic exposure increased with increasing dose in a greater than dose proportional manner (Table 1 and Fig 2). Based on mean AUC336 values, little or no accumulation occurred between Days 1 and 15. In addition, there were no consistent sex-

**Table 1. Toxicokinetic parameters for PF-07209960 in cynomolgus monkeys.**

| Dose | Study Day | $C_{max}$ | AUCt |
|---|---|---|---|
| mg/kg/dose | | (μg/mL) | (μg•h/mL) |
| 0.1 | 1 | 0.319 | 15.1 |
| | 15 | 0.343 | 10.6 |
| | 29[a] | 0.166 | 5.54 |
| 0.3 | 1 | 1.57 | 73.0 |
| | 15 | 1.51 | 54.3 |
| | 29 | 1.15 | 37.9 |
| 1 | 1[b] | 8.10 | 308 |
| | 15[c] | 4.33 | 170 |
| | 29[c] | 5.44 | 174 |
| 3 | 1 | 23.7 | 1060 |

Notes: 3 animals/sex/dose group unless otherwise footnoted.

a. 2 males and 1 female.

b. 3 animals/sex/dose group for $C_{max}$; 1 male and 2 females for $AUC_{336}$.

c. 2 females.

related differences in systemic exposure. The incidence of ADA induction to PF-07209960 was 100% (6 of 6), 100% (6 of 6), 100% (6 of 6), and 67% (4 of 6) for animals administered 0.1, 0.3, 1 or 3 mg/kg/dose, respectively: with an overall incidence rate of 92% (22/24) across all dose groups.

## Receptor (PD-1) occupancy and STAT5 phosphorylation

Following each dosing, nearly 100% of cell surface PD-1 were engaged by PF-07209960 within 15 minutes (IV route) or 6 hours (SC route) at all dose levels, respectively. Four days postdose, the majority of PD-1 on CD8+ T cell surface became free of PF-07209960 and remained drug-free until the next dose. This RO kinetics was similar between dose 1 and dose 2 (Fig 1C and 1D) and was consistent with the PK data (Fig 1A and 1B).

STAT5 phosphorylation is one of the key downstream signaling events of IL-15 receptor activation. We measured phosphorylated STAT5 (pSTAT5) in T cells and NK cells in PF-07209960-dosed monkeys by flow cytometry. Following IV dosing at 0.01 and 0.03 mg/kg, greater percentages of CD8+ T cells became pSTAT5+ than those of CD4+ T cells, followed by NK cells (Fig 1F–1I). This is likely due to higher expression of PD-1 on CD8+ T cells than CD4+ T cells and NK cells (Fig 1E), since the IL-15 activity of PF-07209960 positively correlates with cell surface PD-1 density [21].

## Activation and expansion of T cell and NK cell subsets

In the GLP toxicity study, peripheral blood immunophenotyping data are presented in S4 Table in S1 File and Fig 3. For animals administered ≥0.1 mg/kg/dose, decreases in the numbers of T cells, CD4+ T cells, CD8+ T cells, NK cells, and gdT cells (0.02x-0.34x) were noted on Days 3 and/or 31 of the dosing phase. In general, the decreases were of smaller magnitude in animals administered 0.1 mg/kg/dose but no clear dose relationships were noted. These decreases in lymphocytes were transient and followed by increases in the absolute numbers of these cells. Increases in the numbers of T cells, CD4+ T cells, CD8+ T cells, gdT cells, NK cells, and Treg were noted in animals administered ≥0.1 mg/kg/dose. These increases were generally largest on Day 8 or 22. The increases in gdT cells were of the greatest magnitude, followed

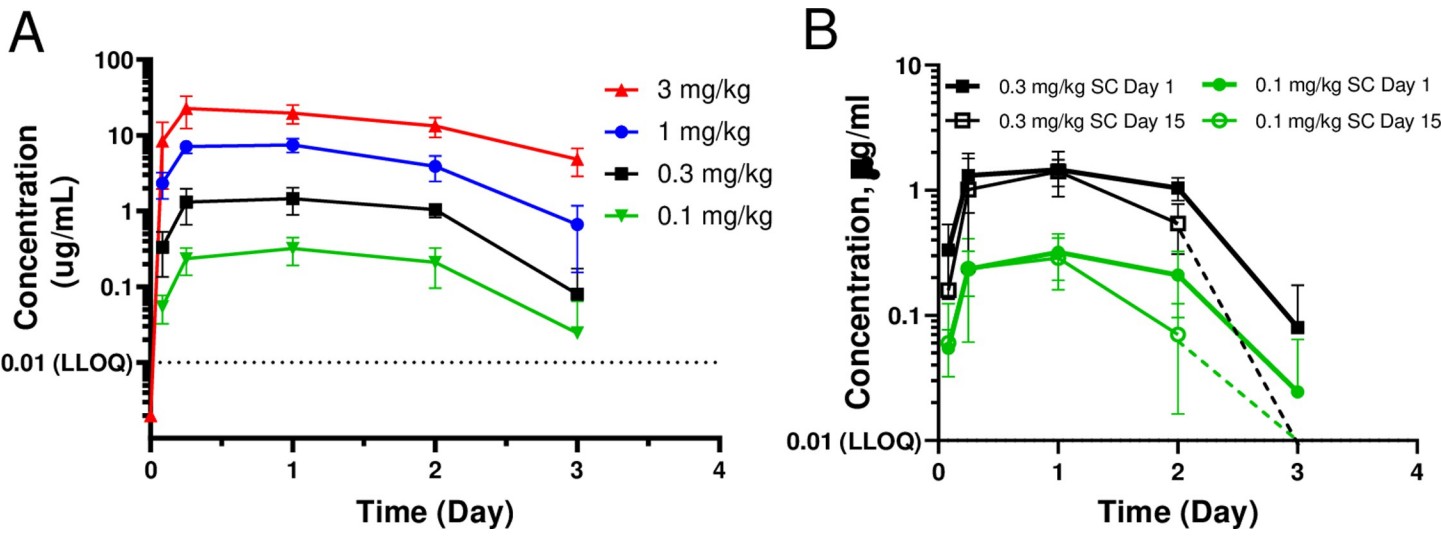

**Fig 2. Pharmacokinetics of PF-07209960 in the GLP tox study.** Cynomolgus monkeys (3 males and 3 females per dose group) were administered PF-07209960 at 0 (vehicle control), 0.1, 0.3, 1, or 3 mg/kg/dose on Days 1, 15, and 29. **A** shows the pharmacokinetics of PF-07209960 following the first dose at 0.1 (green), 0.3 (black), 1 (blue), or 3 (red) mg/kg. **B** shows the pharmacokinetics of PF-07209960 following the first dose on Day 1 and second dose on Day 15 at 0.1 mg/kg/dose (red) or 0.3 mg/kg/dose (blue). Data represent mean and standard deviations from 3 male and 3 female cynomolgus monkeys.

by T cells. Increases in the percentages were observed for CD69+ (activated) CD4+ and CD8 + T cells at ≥0.1 mg/kg/dose. These increases in activated T cells were usually largest on Day 3 or 31. Increases in the percentages of proliferating (Ki-67+) CD4+ T cells and CD8+ T cells were observed in animals administered ≥0.1 mg/kg/dose. The increases in these Ki-67+ T cells generally peaked on Day 8 or 22, which was consistent with the timing of the peak increase in numbers of these T lymphocyte subsets.

## Release of proinflammatory cytokines

Upon activation, T cells and NK cells are expected to produce cytokines. There were dose-dependent increases in serum cytokines IL-6, IL-10, and IFN-γ following the first dose, with substantial increases observed at ≥1 mg/kg (Fig 4). These increases generally peaked on Day 4 (72 hr postdose) and resolved at 96 hours post the Day 1 dose in some animals administered 1 mg/kg/dose but remained elevated in animals that were euthanized on Day 6 of the dosing phase. Increases in IL-6, IL-10, and IFN-γ were not consistently noted in all animals administered 0.1 or 0.3 mg/kg/dose and were minor in magnitude.

## Toxicity of PF-07209960 in cynomolgus monkeys

**Toleration.** PF-07209960-related mortality occurred between Days 4 and 16 of the dosing phase for animals administered 1 or 3 mg/kg/dose on Day 1. All animals (6 in total) administered 3 mg/kg/dose were euthanized in a moribund condition on Day 4 of the dosing phase, with clinical observations that included increased body temperature, liquid and/or nonformed feces, hunched posture, recumbent posture, and decreased general activity. In the 1 mg/kg/dose group, 4 out of the 6 animals underwent unscheduled euthanasia. On Day 6, 2 males and 1 female administered 1 mg/kg/dose were euthanized in a moribund condition, with clinical observations that included liquid feces, vomitus, abnormal skin color consistent with petechiae. In addition to the multifocal petechiae, veterinary observations for 2 of the 3 animals euthanized on Day 6 also included epistaxis, and/or gingival bleeding; these observations were

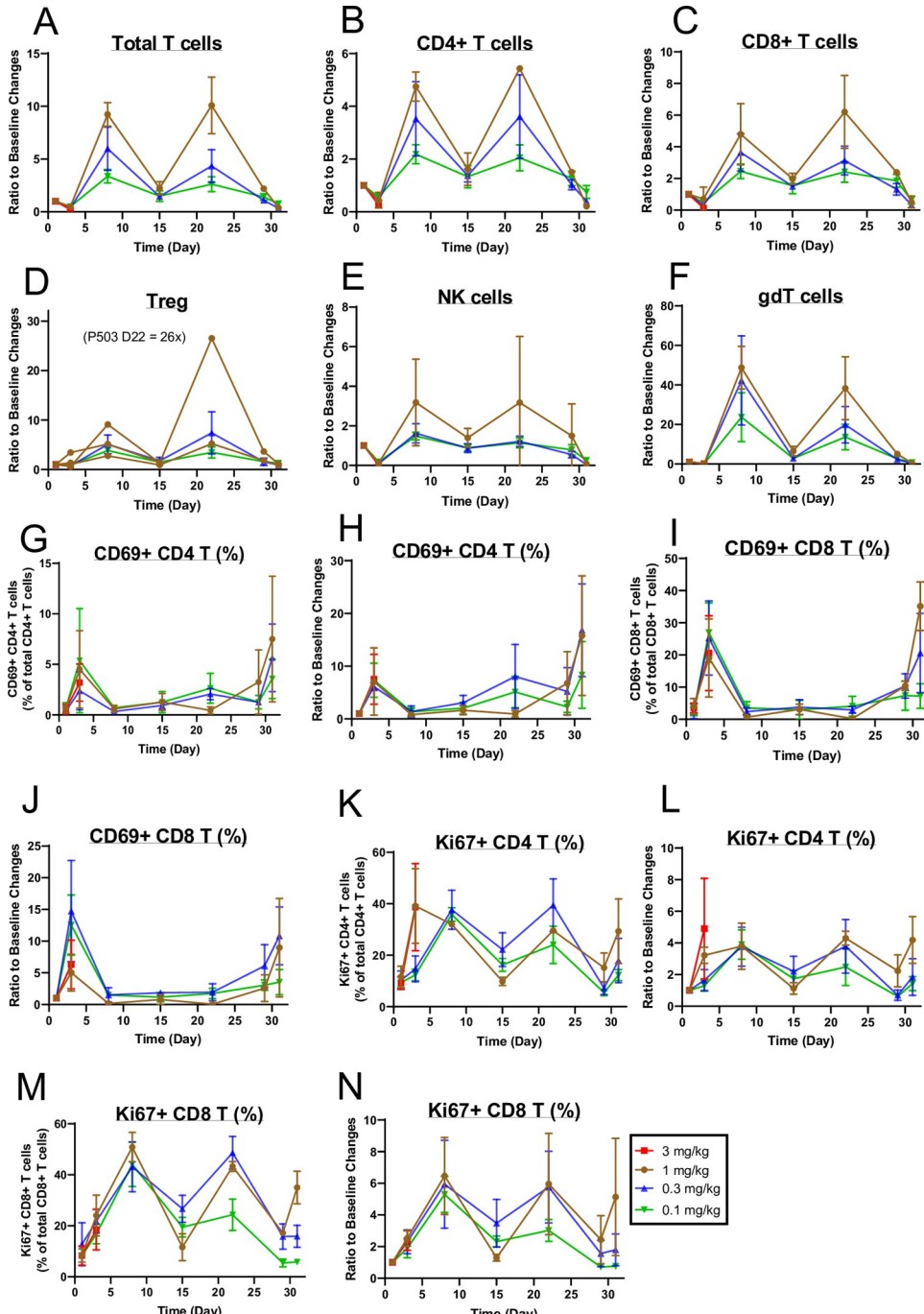

**Fig 3. Immunophenotyping of blood lymphocytes in the GLP tox study.** Cynomolgus monkeys (3 males and 3 females per dose group) were administered PF-07209960 at 0 (vehicle control), 0.1, 0.3, 1, or 3 mg/kg/dose on Days 1, 15, and 29. All animals were euthanized on Day 31. **A-F** show the changes from baseline values (Day 1 predose) in the absolute counts of total T cells (A), CD4+ T cells (B), CD8+ T cells (C), Treg (D), NK cells (E), and gdT cells (F). **G** and **H** show the changes in the percentage of CD69+ CD4+ T cells of the total CD4+ T cells. The percentage values are shown in **G** and the ratio (fold change) from baseline values (Day 1 predose) of the percentage values are shown in **H**. **I** and **J** show the changes in the percentage of CD69+ CD8+ T cells of the total CD8+ T cells. The percentage values are shown in **I** and the ratio (fold change) from baseline values (Day 1 predose) of the percentage values are shown in **J**. **K** and **L** show the changes in the percentage of Ki67+ CD4+ T cells of the total CD4+ T cells. The percentage values are shown in **K** and the ratio (fold change) from baseline values (Day 1 predose) of the percentage values are shown in **L**. **M** and **N** show the changes in the percentage of Ki67+ CD8+ T cells of the total CD8+ T cells. The percentage values are shown in **M** and the ratio (fold change) from baseline values (Day 1 predose) of the percentage values are shown in

**N**. Treg = regulatory T cells; gdT = gamma delta T cells. Data represent mean and standard deviations from 3 males and 3 females. Vehicle control group immunophenotyping data are not plotted. Due to unscheduled euthanasia, no immunophenotyping data were collected from animals dosed at 3 mg/kg/dose beyond Day 3, or from 3 animals beyond Day 3 (P0101, P0102, and P0502) and 1 animal (P0103) beyond Day 15 in the 1 mg/kg/dose group.

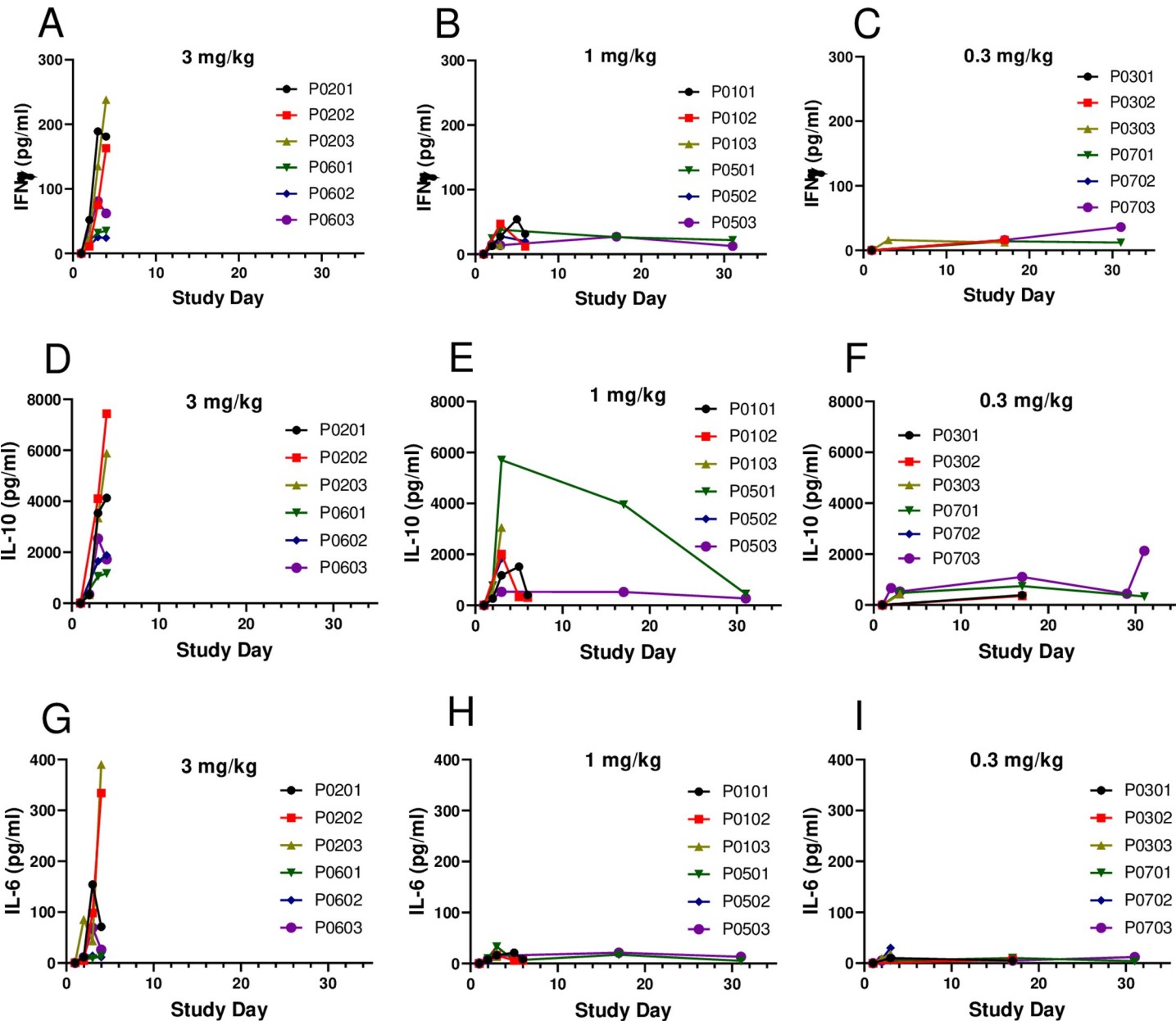

**Fig 4. Serum cytokines in the GLP tox study.** Cynomolgus monkeys (3 males and 3 females per dose group) were administered PF-07209960 at 0 (vehicle control), 0.1, 0.3, 1, or 3 mg/kg/dose on Days 1 and 15. Individual animal serum cytokine IFNγ (**A-C**), IL-10 (**D-F**), and IL-6 (**G-I**) concentrations (μg/mL) for the 3 mg/kg/dose (**A, D, G**), 1 mg/kg/dose (**B, E, H**), and 0.3 mg/kg/dose (**C, F, I**) groups are presented. Male animal identification numbers begin with P01, P02, or P03, and female animal identification numbers begin with P05, P06, or P07. Due to unscheduled euthanasia, no cytokine data were collected from animals dosed at 3 mg/kg/dose beyond Day 4, or from the 4 animals (P0101, P0102, P0103 (males), and P0502 (female)) dosed at 1 mg/kg/dose beyond Day 6.

largely attributed to platelet decreases, described below. Based on continued poor clinical condition, 1 male administered 1 mg/kg/dose was not administered the Day 15 dose and was euthanized on Day 16. The cause of the moribund condition for these animals was undetermined based on clinical or anatomic pathology findings; however, elevated serum proinflammatory cytokines, low platelet counts (discussed below), and mononuclear gastrointestinal infiltrates (esophagus, stomach, duodenum, jejunum, ileum, cecum, and/or colon; Fig 6) may have contributed to the cause of the moribund condition. Other findings related to PF-07209960 administration included mononuclear infiltrates in other organs (brain, sciatic nerve, kidney, urinary bladder, liver, lung, heart, skeletal muscle, tongue, thyroid, mandibular salivary gland, mammary gland, epididymis, prostate, vagina, skin/subcutis, and/or subcutaneous Injection Sites), adrenal cortex hypertrophy, thymus decreased lymphocytes, increased lymphocytes in the red pulp of the spleen, axillary and mesenteric lymph nodes, and GALT, kidney tubule dilation and degeneration, hepatocyte vacuolation, testis seminiferous tubule degeneration, epididymis single cell necrosis and reduced sperm, bone marrow increased hematopoietic cells and fat atrophy, pancreas acinar cell secretory depletion; these findings did not contribute to the moribundity because of their character and/or limited severity. Most of these findings were present at scheduled euthanasia (discussed below in the Anatomic Pathology section).

**Clinical pathology.** Striking decreases in platelet counts were observed in animals administered ≥0.1 mg/kg/dose, with absolute counts reaching 8 (10e3/uL)) in the previously described male at 1 mg/kg/dose. In remaining animals, decreases in platelet counts were transient with a nadir on Day 6 and rebounded to above baseline by Day 8 and remained at this level until the second dose. Following the second dose, platelet count changes were similar to those following the first dose (Fig 5). Nominal prolongations in prothrombin time (with times up to 13.8 seconds) and activated partial thromboplastin time (with times up to 31.5 seconds) were also observed at ≥0.3 mg/kg/dose. Additional key hematology findings were attributed to inflammation, including increases in total white blood cell count (with absolute counts reaching up to144.7 (10e3/uL)) at various time points composed of sporadic increases in neutrophils (with absolute counts reaching up to 18.48 (10e3/uL); occasionally left shifted to band neutrophils), lymphocytes (with absolute counts reaching up to 136 (10e3/uL); including occasional observations of atypical lymphocytes and/or cytoplasmic vacuoles), as well as monocytes, eosinophils, basophils, and/or large unstained cells at ≥ 0.1 mg/kg/dose. Coagulation and clinical chemistry findings attributed to inflammation were also observed, which included increases in fibrinogen (with values up to 472 mg/dL) and C-reactive protein (with values reaching the upper linearity of the assay, >23.5 mg/dL), along with decreases in albumin in animals administered ≥1 mg/kg/dose. Less commonly observed were sporadic increases in blood urea nitrogen (with values up to 84 mg/dL), and creatinine (with values up to 9.3 mg/dL), in animals administered ≥1 mg/kg/dose.

**Anatomic pathology.** The incidence and severity of PF-07209960-related microscopic findings in unscheduled and scheduled euthanasia animals are listed in S2 Table in S1 File and 3. In animals administered ≥0.1 mg/kg/dose and surviving to terminal euthanasia, there were PF-07209960-related mononuclear infiltrates in the kidney (Fig 6), urinary bladder, liver (Fig 6), gall bladder, brain, pituitary gland, sciatic nerve, lung, trachea, heart, skeletal muscle, tongue, gastrointestinal tract, thyroid gland, salivary gland, mammary gland, skin, injection site, prostate, seminal vesicle, epididymis, and/or vagina. In the kidney and liver, mononuclear infiltrates were associated with increased organ weights. In most tissues, the severity of the mononuclear infiltrates was minimal or mild and in general the incidence and/or severity were dose-dependent. In the kidney of one female administered 1 mg/kg/dose the infiltrates were marked and associated with increased blood urea nitrogen and creatinine. In addition to

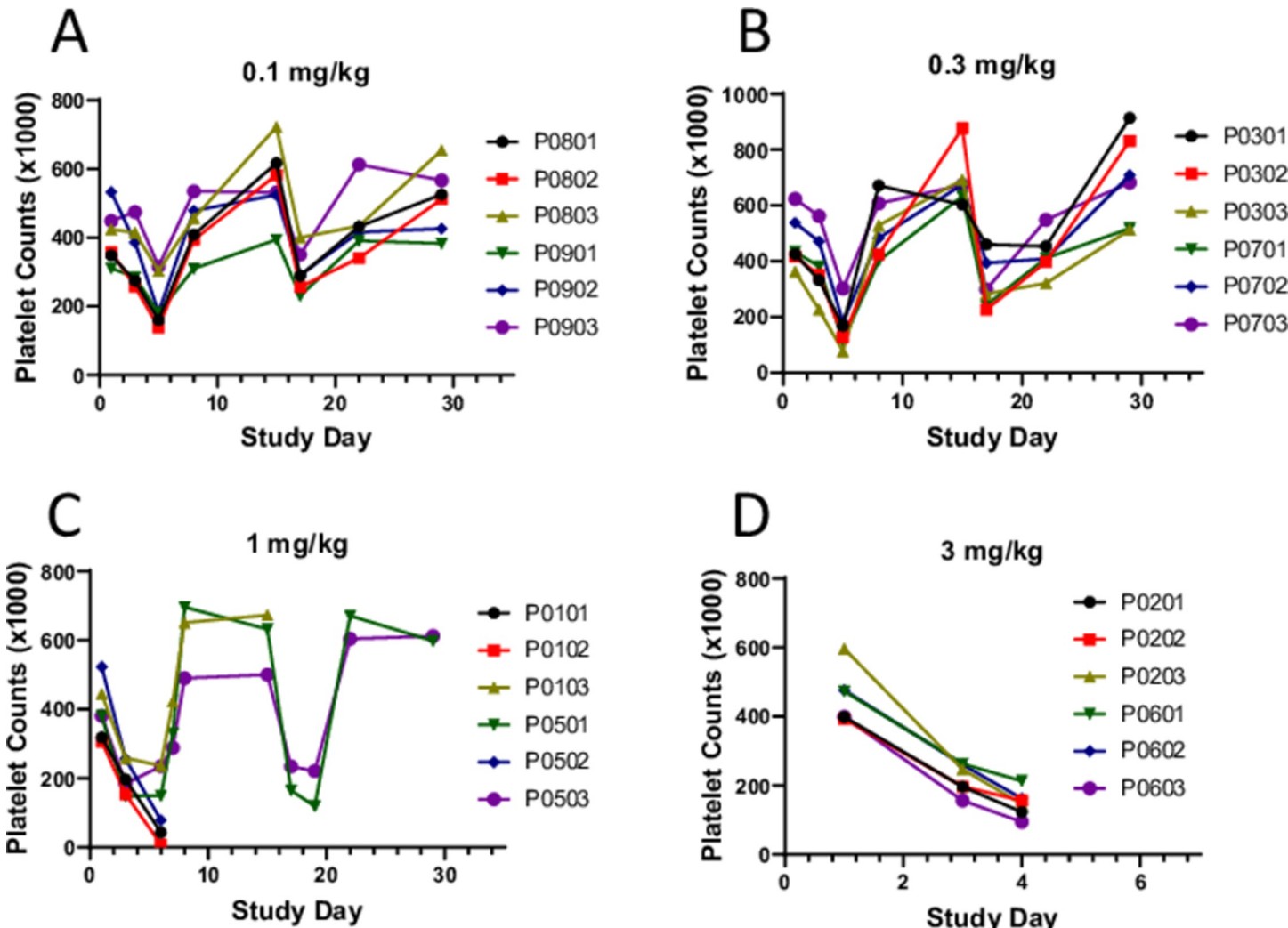

**Fig 5. Platelet count changes in the GLP tox study.** In the GLP tox study, male and female cynomolgus monkeys (3/sex/dose group) were administered PF-07209960 every other week by subcutaneous injections. Individual animal platelet count changes following dosing at 0.1, 0.3, 1, or 3 mg/kg on Day 1 and 15 are shown (**A-D**). **C**. Due to unscheduled euthanasia, no platelet count data are available beyond Day 6 for animals in the 1 mg/kg dose group P0101 (male), P0102 (male), and P0502 (female), and for animal P0103 (male) beyond Day 15. **D**. Due to unscheduled euthanasia, no platelet count data are available for all animals in the 3 mg/kg dose group beyond Day 4.

these mononuclear infiltrates, there were increased lymphocytes in the red pulp of the spleen (correlated with increased spleen weights), and in the mesenteric and/or axillary lymph nodes (Fig 6; correlated with large size macroscopically). These findings generally correlated with clinical pathology findings of increased white blood cell, neutrophil, lymphocyte, monocyte, and leukocyte differential counts, and immunophenotyping finding of increased activation and proliferation of T cells and NK cells in blood (Fig 3).

Other PF-07209960-related findings in animals administered ≥0.1 mg/kg/dose included tubule degeneration in the kidney, single cell necrosis and luminal debris in the epididymis, hypertrophy of the adrenal zona fasciculata (correlated with large size macroscopically and increased weights), increased myeloid/erythroid ratio or hematopoietic cells in the bone marrow, and/or secretory depletion in the pancreas.

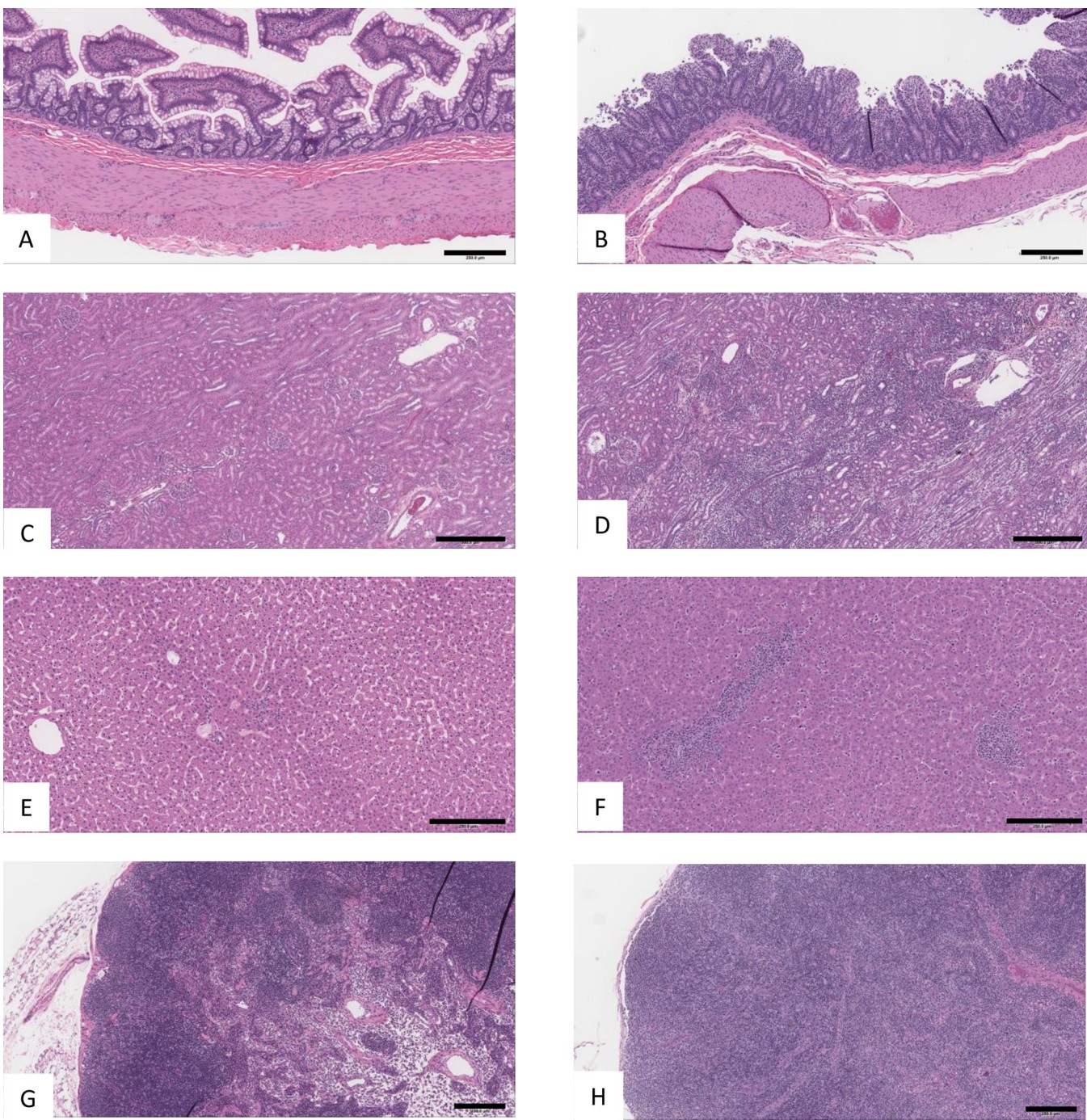

**Fig 6. Representative photomicrographs from GLP toxicology study.** Cynomolgus monkeys (3 males and 3 females per dose group) were administered PF-07209960 at 0 (vehicle control), 0.1, 0.3, 1, or 3 mg/kg/dose on Days 1, 15, and 29. All animals were euthanized on Day 31 as scheduled, except for unscheduled euthanasia of 4 monkeys in the 1 mg/kg dose group (euthanized on Day 6 (3 animals) and 16 (1 animal) and all animals in the 3 mg/kg dose group (euthanized on Day 4). All images are from H&E stained sections. Calibration bar in lower right corner of each image; **A**, **B**, **E**, **F**, **G**, and **H** = 250 um, **C** and **D** = 500 um. Left side of panel (**A**, **C**, **E**, and **G**) are all controls administered 0 mg/kg/dose and euthanized at scheduled euthanasia, they are included for comparison to respective organ sections from PF-07209960-dosed animals (**B**, **D**, F, and **H**) on right. Note the mononuclear infiltrates expanding the lamina propria of the ileum (**B**) from a male euthanized moribund 4 days after receiving a single dose of 3 mg/kg. The degree of infiltrate combined with the villar blunting that is present may have contributed to the moribund condition. At terminal necropsy (images from males administered 0.3 mg/kg/dose), numerous organs had mononuclear infiltrates as illustrated in the kidney (**D**), where the infiltrates surround tubules and expand the interstitium, and liver (**F**). Increased mononuclear cells were also present in lymphoid organs as illustrated by the axillary lymph node (**H**) of a male administered 0.3 mg/kg/dose where the paracortex is expanded by increased lymphocytes.

## Discussion

Cytokines have potent immunomodulating functions and promising utility in immuno-oncology. However, as therapeutic agents, cytokines have some limitations such as poor PK due to small size, undesirable pharmacological effects, and narrow therapeutic index (TI). Various approaches have been described to modify cytokines for cancer immunotherapy to improve these properties [21, 25–30]. One of the attractive approaches of improving PK and TI is to attach cytokines to antibodies that selectively target tumor tissue or tumor infiltrating immune cells [21, 31–33]. Tumor-targeted antibody-cytokine fusions are expected to increase intratumor concentration and efficacy, thereby decreasing the systemic effective dose and toxicity. To facilitate tumor targeting, mutations are introduced into cytokines to reduce their binding affinities to their receptors and their systemic activity [21, 26, 27, 30, 34–39]. We have previously reported an engineered αmPD1-hIL15m antibody-cytokine fusion protein with reduced binding affinity to IL-15 receptors. This molecule showed preferential activation and expansion of intratumor T cells, with greatly reduced activity on peripheral T cells and NK cells [21].

In the current study, we developed an αhPD1-hIL15m molecule (PF-07209960) with similar IL-15 receptor binding properties to αmPD1-hIL15m. In cynomolgus monkeys, PF-07209960 showed extended PK in comparison with recombinant human IL-15 which requires daily dosing [40]. Compared to typical IgG antibody, PF-07209960 had faster clearance [23]; however, its PK profile is similar to other IgG-cytokine fusion proteins [24]. Following the second dose, faster clearance was noticed, which was attributable to expanded lymphocytes pool and ADA. Although ADA was detected in 92% of dosed monkeys, ADA had no or minimal impact on the pharmacological effects (receptor occupancy, STAT5 phosphorylation, and T cell expansion) of PF-07209960 (Figs 1 and 3).

While wild type IL-15 is naturally more potent on NK cells than on T cells [3, 41, 42], PF-07209960 showed higher proliferative activity on T cells than on NK cells (Fig 3A and 3E). This is encouraging since intratumor T cells are more important in IL-15 anti-tumor efficacy [16, 17, 21] and peripheral NK cell activation is associated with IL-15 systemic toxicity [42]. Following administration of PF-07209960, there was a rapid decrease in the numbers of peripheral blood T cells and NK cells (Fig 3). This is consistent with previously reported rhIL-15 in patients [3, 40] and is likely due to redistribution of lymphocytes to various tissues following cell activation. This is also supported by the observed multi-tissue mononuclear cell infiltration from histological evaluation on tissues collected 2 days post the second dose.

One of the interesting observations is that even after PF-07209960 was cleared from circulation, the pharmacological effect (T cell and NK cell proliferation) continued to carry on and reached peak on Day 8. This suggests there was a positive feedback loop of cell activation, cytokine production, and cytokine-driven cell proliferation. After reaching the peak of cell expansion on Day 8, cell numbers began to decline and returned to near baseline level on Day 15. This T cell proliferation kinetics remained the same following the second dose (Fig 3), thus supporting an every other week dosing regimen.

The primary clinical toxicity of IL-15 includes fever, hypotension, platelet decreases, and liver injury [3], which is believed to be associated with the release of proinflammatory cytokines, especially IFN-γ. In the mouse model, Guo et al. showed that the systemic toxicity of IL-15 was mediated by hyperproliferation of activated NK cells and the production of the proinflammatory cytokine IFN-γ [42]. In cynomolgus monkeys, PF-07209960 dose-dependently induced elevation of serum cytokines IFN-γ, IL-6, and IL-10. The increases of these cytokines peaked on Day 3 or 4, coincident with the onset of fever, reduced activity, hunched posture and other clinical signs, suggesting the toxicities of PF-07209960 in monkeys were primarily caused by the release of these cytokines. Similar to the clinical toxicity of rhIL-15, monkeys

administered PF-07209960 also experienced fever and platelet decreases. All monkeys dosed at 3 mg/kg had to be euthanized on Day 4 due to moribund condition. Three monkeys dosed at 1 mg/kg had to be euthanized on Day 6 due to rapid platelet decreases (0.14x, 0.15x, and 0.02x from baseline, Fig 5), accompanied by associated clinical signs including petechiae, epistaxis, and/or gingival bleeding. This combination of findings in these three monkeys was not definitively attributed to disseminated intravascular coagulation (DIC), however. Nominal prolongations of prothrombin time (PT) and/or activated partial thromboplastin time (APTT) were also observed; however, prolongations were not confined to or notably more severe in the individuals that underwent unscheduled euthanasia; fibrinogen was consistently increased (as opposed to decreased, the expected directionality of change with disseminated intravascular coagulation); additionally, there was no microscopic evidence of intravascular coagulation, such as microthrombi in glomerular or pulmonary capillaries. Further complicating clear identification of coagulation cascade activation in these animals was the fact that false APTT prolongations can occur in inflammatory conditions with increased CRP [43]. In remaining animals, the platelet decreases were transient, generally rebounding very quickly from a nadir on Day 6 to counts above baseline level on Day 8. The cause of these decreases was unclear, however, it seemed unlikely that the platelet changes observed in our study were due wholly to peripheral destruction and de novo production. Although megakaryocytes have a high steady-state level of platelet production and can greatly increase production under high demand [44], the extremely rapid rebound in platelet counts observed in our study suggested that the platelet decreases may have been due, at least in part, to a temporary, abnormal distribution of platelets, as this has been described in association with conditions such as acute inflammation, hypothermia, and circulating immune complexes [45–47]. Interestingly, thrombocytopenia that rebounds very shortly after cessation of dosing has also been described in patients undergoing aldesleukin treatment, with peripheral mechanisms such as sequestration and/or reticuloendothelial system clearance proposed as primary causes [48, 49].

At the terminal euthanasia, the kidney finding of mononuclear infiltrates in females administered 1 mg/kg/dose was adverse because of the severity (up to marked) and the association with functional changes observed in clinical chemistry parameters (increased blood urea nitrogen and creatinine). Thus, the no observed adverse effect level (NOAEL) was 0.3 mg/kg/dose. Since the 1 and 3 mg/kg dose levels were not tolerated, the highest non-severely toxic dose (HNSTD) was also 0.3 mg/kg/dose, which corresponded to mean $C_{max}$ and $AUC48$ values of 1.15 µg/mL and 37.9 µg*h/mL, respectively.

In summary, the results of the present study support Phase I clinical evaluation of subcutaneous dosing of PF-07209960 for the treatment of patients with advanced solid tumors.

## Supporting information

**S1 File.**
(XLSX)

## Acknowledgments

We thank Poonam Aggarwal for technical support of PF-07209960 production and binding affinity and PBMC pSTAT4 activity characterizations, Jie Guo for technical support of serum PF-07209960 quantitation, Peter Struss and Brent Kern for technical support of receptor occupancy and pSTAT5 assays, Carol Donovan and Karen Ericson for technical support of blood immunophenotyping, Lisa Dyleski for technical support of PK and ADA analysis, and Charles River and Covance Laboratories for supporting the in vivo animal studies.

## Author Contributions

**Conceptualization:** Changhua Ji.

**Data curation:** Changhua Ji, Bing Kuang, Bernard S. Buetow, Allison Vitsky, Yuanming Xu, Javier Chaparro-Riggers, Eugenia Kraynov.

**Formal analysis:** Changhua Ji, Bing Kuang, Bernard S. Buetow, Allison Vitsky, Yuanming Xu, Javier Chaparro-Riggers.

**Investigation:** Changhua Ji, Bing Kuang, Bernard S. Buetow, Allison Vitsky, Yuanming Xu, Tzu-Hsuan Huang, Javier Chaparro-Riggers, Eugenia Kraynov.

**Methodology:** Bing Kuang, Bernard S. Buetow, Allison Vitsky, Yuanming Xu, Javier Chaparro-Riggers, Eugenia Kraynov.

**Project administration:** Changhua Ji.

**Supervision:** Changhua Ji, Tzu-Hsuan Huang, Eugenia Kraynov, Diane Matsumoto.

**Writing – original draft:** Changhua Ji, Bing Kuang, Bernard S. Buetow, Allison Vitsky, Yuanming Xu, Tzu-Hsuan Huang, Javier Chaparro-Riggers, Eugenia Kraynov, Diane Matsumoto.

**Writing – review & editing:** Changhua Ji, Bing Kuang, Bernard S. Buetow, Allison Vitsky, Yuanming Xu, Tzu-Hsuan Huang, Javier Chaparro-Riggers, Eugenia Kraynov, Diane Matsumoto.

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
