## [Decision Letter · Decision Letter 0]

7 Sep 2023

PONE-D-23-18791Pharmacokinetics, pharmacodynamics, and toxicity of a PD-1-targeted IL-15 in cynomolgus monkeysPLOS ONE

Dear Dr. Ji,

Thank you for submitting your manuscript to PLOS ONE. After careful consideration, we feel that it has merit but does not fully meet PLOS ONE’s publication criteria as it currently stands. Therefore, we invite you to submit a revised version of the manuscript that addresses the points raised during the review process.

The referee’s views of your team’s work are favorable, and they have provided points for revision that will bring the paper to publishable standard. None of the requested revisions require additional experimentation, however you should address all comments adequately in a revised submission. Specifically, I wish to emphasize that substantial areas of improvement include additional literature synthesis/contextualization, additional synthesis and explanation of certain results (see both reviewers’ specific questions on this), specific explanation of data management and access (which ideally should follow FAIRsharing principles, https://journals.plos.org/plosone/s/recommended-repositories, unless adequate justification is provided otherwise), and more standardized and extensive reporting of animal experimental parameters (e.g., the ARRIVE guidelines mentioned by reviewer #1 found here: https://arriveguidelines.org/arrive-guidelines and here: https://journals.plos.org/plosbiology/article?id=10.1371/journal.pbio.3000410).

We look forward to receiving your revised manuscript.

Kind regards,

Nicholas A. Pullen, Ph.D.

Academic Editor

PLOS ONE

Journal Requirements:

3. In order to comply with PLOS ONE's guidelines for non-human primate experiments (http://journals.plos.org/plosone/s/submission-guidelines#loc-non-human-primates), please provide additional details regarding all relevant steps taken to alleviate suffering (anesthesia, analgesia, details about humane endpoints, euthanasia, etc.). Also indicate how often animal care staff monitored the health and well-being of the animals and the criteria used to make such assessments, including specific protocols and thresholds used. Please also list the number of animals that died before euthanasia, the number of animals euthanised before the end of the experiment, and the number euthanised at the end of the experiment.

Reviewers' comments:

Reviewer's Responses to Questions

**Comments to the Author**

1. Is the manuscript technically sound, and do the data support the conclusions?

Reviewer #1: Yes

Reviewer #2: Yes

2. Has the statistical analysis been performed appropriately and rigorously? 

Reviewer #1: No

Reviewer #2: Yes

3. Have the authors made all data underlying the findings in their manuscript fully available?

Reviewer #1: No

Reviewer #2: Yes

4. Is the manuscript presented in an intelligible fashion and written in standard English?

Reviewer #1: Yes

Reviewer #2: Yes

5. Review Comments to the Author

Reviewer #1: General Comments

This is an interesting and potentially important paper from an industry team, which clearly represents substantial work. The literature context is nicely described, and the interpretation of much of the data is reasonable. However, more detail is needed in several areas.

Specific Comments

Animal care, human criteria for euthanasia, and euthanasia drugs/details should be provided. The authors should apply the ARRIVE guidelines for reporting of animal experiments.

The basis for the numbers of animals used should be stated. Statistical justification may not be possible with such a low N, but a rationale or guideline could be given.

The data availability statement is not acceptable; specific details are required. The authors are encouraged to read the instructions.

Lines 6 and 7: Abbreviations should be spelled out.

Line 31: The range for T1/2 is broad, 8-13 hours; suggest providing a mean.

Lines 99-123 and elsewhere in the methods: Please define ambiguous abbreviations on first use; some are defined later in the paper or not at all.

Lines 125-132, 178-179, Tables, Figures: More details are needed regarding the animals.

How many animals were used? There are different numbers in different parts of the paper, with some information emerging only in the figure legends. Please clarify total number of males and females and age of animals used in each study up front in the methods.

What was the genetic strain and provenance of the animals?

Were the animals treatment naïve? If they were returned to the colony after this study, does that mean they might have been in prior studies of antibody-like drugs? If so they might have pre-existing ADA to similar agents.

Lines 171-175: Please use past tense if the work was actually conducted as described. Please remove the use of infinitive or future tense.

Lines 176-177: The statement “(that do not require analyses)” is unclear, please explain or remove.

Lines 180-183: Please describe the frequency of animal health assessments.

Lines 183-184: Please describe the timing and frequency of blood collections for PK and ADA analysis.

Line 186: Please provide the dose of pentobarbital used.

Line 188: “may include” is vague, please just give the criteria that were actually used.

Lines 196-198 and 392-404: Please provide a list of organs that were were examined. Were any tissues unaffected? Were there differences by dose? What were the findings in animals that did not survive until day 31? Were all animals affected identically? A table would be helpful.

Line 217: How long were cells refrigerated before flow cytometry analysis? Please specify at least an upper limit, “less than ___”.

Line 234, 285 and elsewhere: Please correct the document for errors in English grammar, for example, the appropriate use of articles “a” “an” “the”, etc.

Table 1: Footnotes are excessive; please put the information about sex and time in the body of the table.

Line 320-321: This sentence needs clarification. Does 6 hours refer to the SQ route?

Line 375-391: A statement of fold change does not provide enough information. For example, an 11-fold increase in neutrophil count is noteworthy but it is not possible to interpret without knowing “11 fold of what”; similarly for platelets and clinical chemistry. Please provide absolute and differential counts.

Line 384: Were the animals septic?

Figure 1E: Please specify the day of measurement.

Line 533: How many animals underwent unscheduled euthanasia for illness in the 1 mg/kg group?

Figure 5: The authors suggest that thrombocytopenia was a transient sequestration effect; this might be true at the low doses, but why was there such profound thrombocytopenia in the animals that succumbed? Was there possibly a consumptive coagulopathy, or an anti-platelet effect of the drug? Please show data from all groups.

Reviewer #2: The IL-15-PD-1 molecule produced by the investigators is exciting and important for the field. It represents a shift in drug development to bispecific molecules. The current study is a follow-up from a prior preclinical study.

Similar to other IL-15 agents, the study drug induced expansion and KI67 of lymphocyte subsets in the periphery. Do these lymphocyte subsets express PD-1? If not, is the study drug acting in a PD-1 dependent manner? Is it possible that reduced dosing would target PD-1hi cells without PD1neg lymphocytes?

Figure 3 reports fold change. Most studies of this nature show the percentage of the marker under evaluation or the percentage of a particular lymphocyte population. Providing data graphed in this manner would be helpful for comparison purposes. Also, can the authors provide their data on NK cell KI67 expression? This seems relevant given the authors argument that NK cells will not be targeted as efficiently.

The authors compare their drug to other monkey studies with IL-15. It may be useful to compare results with other IL-15 studies where IL-15 is linked to an Fc. Rhode et al report in Cancer Immunology Research in 2016 “Comparison of the super agonist complex, ALT-803, to IL-15-“

Is there prior literature about platelets and high dose IL-2 that may be useful for the authors observations? Oleksowicz – Dutcher in 1994 may be useful.

The authors write in the methods that samples were refrigerated until flow cytometry analysis. What is the range and the maximal time before flow cytometry?

6. PLOS authors have the option to publish the peer review history of their article (what does this mean?). If published, this will include your full peer review and any attached files.

Reviewer #1: No

Reviewer #2: No

---

## [Author Response · Author response to Decision Letter 0]

30 Oct 2023

Editor's comments are addressed in the revised cover letter.

Reviewer #1: General Comments

This is an interesting and potentially important paper from an industry team, which clearly represents substantial work. The literature context is nicely described, and the interpretation of much of the data is reasonable. However, more detail is needed in several areas.

Specific Comments

Animal care, human criteria for euthanasia, and euthanasia drugs/details should be provided. The authors should apply the ARRIVE guidelines for reporting of animal experiments.

Author Response: we have added some additional information to these monkey studies.

The basis for the numbers of animals used should be stated. Statistical justification may not be possible with such a low N, but a rationale or guideline could be given. 

Author Response: The number of monkeys used in the IND-enabling GLP tox study was following regulatory guidelines (ICH S9). We added a sentence on Page 8 under “Cynomolgus monkey GLP toxicity study”.

The data availability statement is not acceptable; specific details are required. The authors are encouraged to read the instructions. 

Author Response: Animal study related information are stored at Pfizer database and additional information beyond those in the manuscript can be provided upon request to the corresponding author.

Lines 6 and 7: Abbreviations should be spelled out. 

Author Response: they were spelled out.

Line 31: The range for T1/2 is broad, 8-13 hours; suggest providing a mean. 

Author Response: We modified the sentence to: “PF-07209960 showed dose dependent pharmacokinetics with a terminal T1/2 of 8 and 13 hours following IV administration at 0.03 and 0.1 mg/kg, respectively. The clearance is faster than a typical IgG1 antibody”

Lines 99-123 and elsewhere in the methods: Please define ambiguous abbreviations on first use; some are defined later in the paper or not at all. 

Author Response: Thanks for this suggestion, we added the definitions of these abbreviations.

Lines 125-132, 178-179, Tables, Figures: More details are needed regarding the animals. 

How many animals were used? There are different numbers in different parts of the paper, with some information emerging only in the figure legends. Please clarify total number of males and females and age of animals used in each study up front in the methods. 

Author Response: Thanks reviewer for this suggestion. In this manuscript, data from two separate monkey studies are used: PKPD study and the GLP tox study. We added the number of animals and other information to these two studies in the M&M section. 

What was the genetic strain and provenance of the animals?

Were the animals treatment naïve? If they were returned to the colony after this study, does that mean they might have been in prior studies of antibody-like drugs? If so they might have pre-existing ADA to similar agents. 

Author Response: All monkeys used in this study are naïve monkeys. We also added the monkey origin, BW, and age information to the PKPD study and GLP tox study sections in M&M.

Lines 171-175: Please use past tense if the work was actually conducted as described. Please remove the use of infinitive or future tense. 

Author Response: Thanks for pointing this out. We made the appropriate changes.

Lines 176-177: The statement “(that do not require analyses)” is unclear, please explain or remove. 

Author Response: Good point, we deleted it.

Lines 180-183: Please describe the frequency of animal health assessments. 

Author Response: we Added these information.

Lines 183-184: Please describe the timing and frequency of blood collections for PK and ADA analysis. 

Author Response: We added the timepoints.

Line 186: Please provide the dose of pentobarbital used. 

Author Response: 50 mg/kg, added this to the updated manuscript.

Line 188: “may include” is vague, please just give the criteria that were actually used. 

Author Response: We made the modification. Deleted “may”. These are the general guidelines of unscheduled euthanasia. Under which, we also described the specific reasons for the authorization of the animals in our study.

Lines 196-198 and 392-404: Please provide a list of organs that were were examined. Were any tissues unaffected? Were there differences by dose? What were the findings in animals that did not survive until day 31? Were all animals affected identically? A table would be helpful. 

Author Response: The list of all organs/tissues examined and the microscopic examination findings (unscheduled and scheduled) in the GLP toxicity study have been added as supplemental tables. We also added more detailed information about the unscheduled euthanasia in the Result section. In general, histological findings are similar in these unscheduled euthanasia animals, however, since the mortality is caused by cytokine release, individual animal differences in tissue histological change presentations are expected. 

Line 217: How long were cells refrigerated before flow cytometry analysis? Please specify at least an upper limit, “less than ___”. 

Author Response: We added this information (less than 72 hours) in the updated manuscript.

Line 234, 285 and elsewhere: Please correct the document for errors in English grammar, for example, the appropriate use of articles “a” “an” “the”, etc. 

Author Response: Thanks the reviewer for pointing these out. Corrections have been made in the updated version.

Table 1: Footnotes are excessive; please put the information about sex and time in the body of the table. Author Response: Thanks for the suggestion. We can removed the AUC time footnotes, since the column title is AUCt and we don’t have to specify the time for the last measurable concentration in each dose group in the table. We leave the animal # and sex in footnote. 

Line 320-321: This sentence needs clarification. Does 6 hours refer to the SQ route? 

Author Response: Thanks the reviewer for identifying this error. We made the correction below: “Following each dosing, nearly 100% of cell surface PD-1 were engaged by PF-07209960 within 15 minutes (IV route) or 6 hours (SC route) at all dose levels, respectively.”

Line 375-391: A statement of fold change does not provide enough information. For example, an 11-fold increase in neutrophil count is noteworthy but it is not possible to interpret without knowing “11 fold of what”; similarly for platelets and clinical chemistry. Please provide absolute and differential counts. 

Author Response: This section has been updated to include the most extreme absolute value for key parameters 

Line 384: Were the animals septic? 

Author Response: No, none of the animals were septic.

Figure 1E: Please specify the day of measurement. 

Author Response: measured on Day 1 predose, added this information to the figure legend. 

Line 533: How many animals underwent unscheduled euthanasia for illness in the 1 mg/kg group? 

Author Response: 4 in total. 

Figure 5: The authors suggest that thrombocytopenia was a transient sequestration effect; this might be true at the low doses, but why was there such profound thrombocytopenia in the animals that succumbed? Was there possibly a consumptive coagulopathy, or an anti-platelet effect of the drug? Please show data from all groups. 

Author Response: 

Graphs have been updated to include data from all groups. 

There was evidence of activation of the coagulation cascade in some of the animals that underwent unscheduled euthanasia, as indicated by prolongation of prothrombin time (PT) and activated partial thromboplastin time (APTT); however, prolongations were not confined to or notably more severe in the individual animals with the most profound thrombocytopenia. Additionally, there were consistent increases rather than decreases in fibrinogen, and there was no microscopic evidence of disseminated intravascular coagulation. Further complicating clear identification of coagulation cascade activation in these animals is the fact that false APTT prolongations can occur in inflammatory conditions (ref below). However, it is certainly possible there was an additional or alternate effect other than sequestration on platelets, especially in unscheduled euthanasia animals; the Discussion section has been updated to reflect this.

(van Rossum AP, Vlasveld LT, van den Hoven LJ, de Wit CW, Castel A. False prolongation of the activated partial thromboplastin time (aPTT) in inflammatory patients: interference of C-reactive protein. Br J Haematol. 2012 May;157(3):394-5.)

Reviewer #2: The IL-15-PD-1 molecule produced by the investigators is exciting and important for the field. It represents a shift in drug development to bispecific molecules. The current study is a follow-up from a prior preclinical study. 

Similar to other IL-15 agents, the study drug induced expansion and KI67 of lymphocyte subsets in the periphery. Do these lymphocyte subsets express PD-1? If not, is the study drug acting in a PD-1 dependent manner? Is it possible that reduced dosing would target PD-1hi cells without PD1neg lymphocytes? _

Author Response: This is a great comment. Unfortunately, we didn’t measure PD1 expression or PD1+ cell % change kinetics in the GLP tox study. We have shown in our previous publication that our drug has higher activity on T cells that express PD1 (Xu Y, et al. An Engineered IL15 Cytokine Mutein Fused to an Anti-PD1 Improves Intratumoral T-cell Function and Antitumor Immunity. Cancer Immunol Res 2021;9(10):1141-57). We also think that by reducing dose, we probably can see lower activity on PD1- T cells in peripheral blood or tissues while maintain sufficient activity on tumor infiltrating T cells that express higher PD1. 

Figure 3 reports fold change. Most studies of this nature show the percentage of the marker under evaluation or the percentage of a particular lymphocyte population. Providing data graphed in this manner would be helpful for comparison purposes. Also, can the authors provide their data on NK cell KI67 expression? This seems relevant given the authors argument that NK cells will not be targeted as efficiently. 

Author Response: Unfortunately, we didn’t analyze Ki67+ NK cells in this study. For the lymphocyte subsets, the effect on them by PF-07209960 is expansion/proliferation, and the best way of showing this is the change of cell numbers (counts) from baseline values (fold change). There is little if any effects on the percentages of these subsets. We agree with the reviewer that adding graphs of the percentages of Ki-67 and CD69 positive T cells can provide additional information. We added these graphs to Fig 3.

The authors compare their drug to other monkey studies with IL-15. It may be useful to compare results with other IL-15 studies where IL-15 is linked to an Fc. Rhode et al report in Cancer Immunology Research in 2016 “Comparison of the super agonist complex, ALT-803, to IL-15-“ 

Author Response: This is a great question. Since this is a drug development program and we need to be focused on our compound PF-07209960 and generate minimal in vivo toxicity study data that meet regulatory requirement. Therefore we didn’t include competitor’s IL-15 molecules in our toxicity studies. However, our data is generally consistent with Rhode’s publication with some differences (eg. PF-07209960 showed less potent proliferative effect on NK cells compared with T cells, while ALT-803 has the opposite effect, more potent pro-proliferative effect on NK cells than on T cells). 

Is there prior literature about platelets and high dose IL-2 that may be useful for the authors observations? Oleksowicz – Dutcher in 1994 may be useful. 

Author Response: Although the effects we saw were unique in some ways from the effects described in the literature in association with IL-2 administration, it is possible there were other causes/contributors to the platelet decreases observed in this study. The Discussion section has been updated.

The authors write in the methods that samples were refrigerated until flow cytometry analysis. What is the range and the maximal time before flow cytometry? 

Author Response: less than 72 hours before being analyzed on a flow cytometer. We added this information to the Fig legend.

---

## [Decision Letter · Decision Letter 1]

3 Jan 2024

PONE-D-23-18791R1Pharmacokinetics, pharmacodynamics, and toxicity of a PD-1-targeted IL-15 in cynomolgus monkeysPLOS ONE

Dear Dr. Ji,

Thank you for submitting your manuscript to PLOS ONE. After careful consideration, we feel that it has merit but does not fully meet PLOS ONE’s publication criteria as it currently stands. Therefore, we invite you to submit a revised version of the manuscript that addresses the points raised during the review process.

We look forward to receiving your revised manuscript.

Kind regards,

Nicholas A. Pullen, Ph.D.

Academic Editor

PLOS ONE

***Request from the Editorial Staff:

We noted that you declared in your Data Statement the following:

"All relevant data are within the manuscript and its Supporting Information files. Additional data can be requested by contacting the corresponding author (pending institutional legal approval)."

Please note that all PLOS journals require that authors adhere to our policies for sharing of data and materials: http://journals.plos.org/plosone/s/data-availability. We do not consider manuscripts for which the underlying data cannot be shared. PLOS’ data policy requires underlying data to be deposited to a data repository unless the data are subject to ethical restrictions or owned by someone other than the authors (https://journals.plos.org/plosone/s/data-availability#loc-acceptable-data-access-restrictions).

To comply with our data availability policies, can you please clarify if you are able to provide the minimum dataset required to reproduce the results of this study. In addition, we would ask that you provide all contact details for where an interested researcher would need to apply to gain access to the relevant data. Please note that it is not acceptable for an author to be the sole named individual responsible for ensuring data access. If the minimum dataset data cannot be publicly available with the publication, unfortunately, we will not be able to further proceed with your submission.

We appreciate your attention to these requests and look forward to your response.

Journal Requirements:

Reviewers' comments:

Reviewer's Responses to Questions

**Comments to the Author**

1. If the authors have adequately addressed your comments raised in a previous round of review and you feel that this manuscript is now acceptable for publication, you may indicate that here to bypass the “Comments to the Author” section, enter your conflict of interest statement in the “Confidential to Editor” section, and submit your "Accept" recommendation.

Reviewer #1: All comments have been addressed

2. Is the manuscript technically sound, and do the data support the conclusions?

Reviewer #1: Yes

3. Has the statistical analysis been performed appropriately and rigorously? 

Reviewer #1: Yes

4. Have the authors made all data underlying the findings in their manuscript fully available?

Reviewer #1: No

5. Is the manuscript presented in an intelligible fashion and written in standard English?

Reviewer #1: Yes

6. Review Comments to the Author

Reviewer #1: The authors have adequately addressed my prior comments. I have no additional comments. The restriction of raw data availability causes some minor concern, but I defer to the editors on that point.

7. PLOS authors have the option to publish the peer review history of their article (what does this mean?). If published, this will include your full peer review and any attached files.

Reviewer #1: No

---

## [Author Response · Author response to Decision Letter 1]

15 Jan 2024

Dear Editor Dr. Nicholas A. Pullen,

Thank you and the reviewers for reviewing the revised manuscript and positive comments! 

To further address the request for raw data, we have provided the individual animal blood immunophenotyping results in Supplemental Table 4. Here we resubmit the revised manuscript with the additional Supplemental Table 4 for your review. We added a sentence in the Result section, in the beginning of “Activation and Expansion of T Cell and NK Cell Subsets” as follows: In the GLP toxicity study, peripheral blood immunophenotyping data are presented in Supplemental Table 4 and Fig. 3 (page 17, Lines 356-357). 

Kind regards,

Changhua Ji

---

## [Editor Report · Decision Letter 2]

22 Jan 2024

Pharmacokinetics, pharmacodynamics, and toxicity of a PD-1-targeted IL-15 in cynomolgus monkeys

PONE-D-23-18791R2

Dear Dr. Ji,

We’re pleased to inform you that your manuscript has been judged scientifically suitable for publication and will be formally accepted for publication once it meets all outstanding technical requirements.

Sincerely,

Nicholas A. Pullen, Ph.D.

Academic Editor

PLOS ONE

Additional Editor Comments (optional):

Thank you for your willingness to meet the data sharing expectations of the journal.
---

## [Editor Report · Acceptance letter]

25 Jan 2024

PONE-D-23-18791R2 

PLOS ONE

Dear Dr. Ji, 

I'm pleased to inform you that your manuscript has been deemed suitable for publication in PLOS ONE. Congratulations! Your manuscript is now being handed over to our production team.

Kind regards, 

on behalf of

Dr. Nicholas A. Pullen 

Academic Editor

PLOS ONE